# Mechanism of spirometry associated gastro-esophageal reflux in individuals undergoing esophageal assessment

**Matthew Xu[1], John D. Brannan[2], Vincent Ho[1,3], Jerry Zhou[1] \***

**1** School of Medicine, Western Sydney University, Campbelltown, NSW, Australia, **2** Department of Respiratory & Sleep Medicine, John Hunter Hospital, New Lambton, NSW, Australia, **3** Department of Gastroenterology, Campbelltown Hospital, Campbelltown, NSW, Australia

\* j.zhou@westernsydney.edu.au

**Data Availability Statement:** All spirometry and manometric raw data are available from the Figshare database (accession number 10.6084) or

## Abstract

Persistent variability observed during spirometry, even when technical and personal factors are controlled, has prompted interest in uncovering its underlying mechanisms. Notably, our prior investigations have unveiled that spirometry has the potential to trigger gastro-esophageal reflux in a susceptible population. This current study embarks on elucidating the intricate mechanisms orchestrating reflux induced by spirometry. To achieve this, we enlisted twenty-four (24) participants exhibiting reflux symptoms for esophageal assessment. These participants underwent two sets of spirometry sessions, interspersed with a 10-minute intermission, during which we closely scrutinized fluid flow dynamics and esophageal function through high-resolution impedance esophageal manometry. Our comprehensive evaluation juxtaposed baseline manometric parameters against their equivalents during the initial spirometry session, the intervening rest period, and the subsequent spirometry session. Remarkably, impedance values, serving as a metric for fluid quantity, exhibited a substantial elevation during each spirometry session and the ensuing recovery interval in the pan-esophageal and hypopharyngeal regions when compared to baseline levels. Additionally, the resting pressure of the lower esophageal sphincter experienced a noteworthy reduction subsequent to the first bout of spirometry (13.6 ± 8.8 mmHg) in comparison to the baseline pressure (22.5 ± 13.3 mmHg). Furthermore, our observations unveiled a decline in spirometric parameters—FEV1 (0.14 ± 0.24 L, P = 0.042) and PEFR (0.67 L/s, P = 0.34)—during the second spirometry session when contrasted with the first session. Collectively, our study underscores the compelling evidence that spirometry maneuvers can elicit gastro-esophageal reflux by eliciting intra-esophageal pressure differentials and inducing temporary relaxation of the lower esophageal sphincter.

## Introduction

Spirometry is the most common pulmonary function test for the diagnosis and monitoring of respiratory disorders. Despite its safety and reproducibility [1], variability in spirometry results

via the following link https://doi.org/10.6084/m9.figshare.24012702.v1

**Funding:** The author(s) received no specific funding for this work.

**Competing interests:** The authors have declared that no competing interests exist.

can persist in some patients despite control for technical factors and patient technique. Recent evidence suggests that spirometry maneuvers, can induce gastroesophageal reflux (GER) [2]. This phenomenon has implications for spirometry interpretation and patient management. We previously reported forced vital capacity (FVC) maneuver provoked GER in 45% of individuals with GER symptoms. Within this subgroup, GER events were observed during both FVC maneuver and within a 10 minute recovery period immediately afterwards. Furthermore, repeat spirometry testing after the recovery period saw a significant decrease in spirometry parameters; peak expiratory flow rate (PEFR) and forced expiratory volume ($FEV_1$).

A previous study document a cough-like expiratory efforts termed "deflation cough" noted during spirometry FVC maneuvers in individuals with GER symptoms [3]. Following on from this, inhibition of deflation cough was demonstrated through pre-treatment with anti-reflux medication [4]. These studies suggest a causative role of acidic GER in spirometry variability. However, the mechanisms linking spirometry to GER and its potential impact on lung function remain unclear.

We aim to determine the mechanism of spirometry induced GER using high-resolution impedance manometry (HRIM). Our primary objectives were to evaluate refluxate dynamics during spirometry to determine possible mechanisms that could predispose the individual to GER during recovery and subsequent spirometry attempts.

## Materials and methods

### Patient selection

The study was approved by the South West Sydney Local Health District Ethics Committee (2020/ETH03098). All participants provided informed consent to take part in this study and the use of their medical record for research. From September 2021 –December 2022, patients aged 18 to 90 years old attending the Gastrointestinal Motility Clinic at Camden Hospital, NSW Australia for a HRIM and ambulatory 24hr pH study were recruited. The participants in this study were representative of a population that experiences frequent GER symptoms and/or are diagnosed with gastro-esophageal reflux disease (GERD). Patients with GERD were defined by having an esophageal pH <4 for ≥6% of total study time and a DeMeester score ≥14.72 during the 24-hour ambulatory pH study [5]. All patients were off proton pump inhibitors for at least 7 days prior to study. Participants were screened for existing diagnosis of any lung disease and medications with known effects on esophageal function.

Exclusion criteria for study participants were; a history of prior gastric or esophageal surgery, history of nasal and eye surgery, a major esophageal motor disorder by Chicago Classification [6] (i.e. achalasia, aperistalsis, jackhammer, distal esophageal spasm), and pregnant and lactating females. Patients who currently used opioid analgesics and anticholinergic drugs were excluded, but patients who were taking a low dose of tricyclic antidepressants, for example nortriptyline (up to 50 mg/d) or amitriptyline (up to 25 mg/d) were eligible to participate provided these medications were commenced 3 months before the study. Patients that failed to complete HRIM or ambulatory pH study were excluded (complete HRIM study criteria consist of at least 10 complete water swallows and complete pH study criteria is at least 24hrs of continuous pH monitoring).

### Esophageal function testing

A solid-state HRIM catheter consisting of 36 circumferential sensors spaced at 1 cm intervals (Medtronic Inc., Shoreview, MN) and 18 impedance sensors spaced at 2 cm intervals was transnasally intubated with the patient in a seated position. After a landmark calibration phase (20–60 seconds), ten 5 mL 0.9% saline swallows spaced 20 to 30 seconds apart were completed.

The salt content of salient solution reduces the impedance level (kΩ), visually represented by purple overlay. Following standard esophageal assessment, another landmark calibration was performed before participants were asked to perform two sets of spirometry maneuvers separated by a 10 minute recovery period while intubated with the HRIM catheter. Thermal compensation was applied at the end of the studies using ManoView ESO 3.0.1 Software (Medtronic Inc., Shoreview, MN). Following HRIM studies, patients with suspected GERD underwent ambulatory pH studies as per standard clinical protocol. A multichannel intraluminal impedance pH catheter (Medtronic Inc., Shoreview, MN) was transnasally intubated with positioning confirmed based on the manometrically identified lower esophageal sphincter (LES). The catheter consisted of a pH sensor positioned 5 cm above the LES. The pH study was analyzed using AccuView 5.2 (Medtronic Inc., Shoreview, MN).

## Spirometry

Spirometry was conducted using Spiroscout (Ganshorn Medizin Electronic, Niederlauer, Germany) and laptop computer in accordance with the American Thoracic Society/European Respiratory Society (ATS/ERS) spirometry protocol [7] with the addition of the intubated impedance high-resolution manometry catheter. Spiroscout has a reported accuracy deviation of 2.3%, -0.145 L and reproducibility of 0.38% for test parameters in healthy individuals [8]. Attempt was made to meet ATS/ERS criteria for each spirometry sets however, this was not achievable in all studies due to the participant's response to the manometry catheter and its impact on performing spirometry. Participants were instructed to inhale to total lung capacity before performing a FEV maneuver and expiring to residual volume, for a minimum of three attempts and no more than five. We recorded the forced vital capacity (FVC), forced expiratory volume in 1 second (FEV1) and peak expiratory flow rate (PEFR). Subjects then had a 10 minute recovery period, before performing a second set of spirometry. Spirometry studies were reviewed using Ganshorn LFX 1.8 (Ganshorn Medizin Electronic, Niederlauer, Germany).

## Data collection

Impedance values were obtained with the "Smart Mouse" selection tool in ManoView ESO software. The pan-esophageal impedance was defined by the segment between the manometrically defined upper esophageal sphincter (UES) and LES. The average pan-esophageal impedance value was calculated using an area 1 cm below the UES and 1 cm above the LES over a specified period of time. We defined the hypopharynx as the area 1 cm above the manometrically defined UES. The average impedance value at hypopharynx was calculated using area 1 cm above the UES over a specific period of time. Baseline impedance and LES pressure values were measured from the average over a 20 second period chosen based on the absence of detected swallows. The pre-spirometry baseline values were measured during a second landmark calibration period immediately before initiation of spirometry study. The post-spirometry recovery values were measured following the 10-minute recovery period after 1st spirometry set. Impedance values were also measured during each FVC maneuver and averaged for each spirometry set.

## Statistical analysis

Continuous data were reported as mean with SD or median with range. Categorical data were summarized as frequencies and percentages. The differences between continuous variables were assessed using analysis of variance for multiple group comparisons or the Kruskal-Wallis test to compare medians among the groups when nonparametric tests were indicate. Student t

test or Wilcoxon tests were used for comparisons between 2 groups. Expiratory Disproportion Index (EDI) was used as an indicator for laryngotracheal stenosis (stenosis EDI > 50); possible response to the presence of spirometry-induced reflux. EDI was calculated using this formula: EDI = $FEV_1$/PEFR × 100) [9]. Statistical analyses were completed using SPSS 25.0 for Windows (SPSS Inc., Chicago, IL, USA).

## Results

### Patient characteristics, manometric, and pH study findings

The patient population (n = 24) was predominantly female with a mean of 54 years of age (SD ±14.7) (**Table 1**). Apart from 4 participants with non-symptomatic asthma, all participants did not have any other underlying lung diseases. There were no current or past smokers in the study cohort. Total of 17 (71%) patients were identified to have GERD following 24hr pH monitoring, while no significant levels of weakly-acidic or non-acidic reflux was detected in any patients. Hiatal hernia was detected in 9 (37.5%) patients, weak resting LES in 5 (21%) patients, and ineffective esophageal motility in 5 (21%) patients, all of which could predispose to reflux pathophysiology.

### Impedance data

Baseline impedance levels in the esophagus and hypopharynx was observed at 2.52 ±0.74 kΩ and 3.59 ±2.20 kΩ, respectively (**Table 2**). Pan-esophageal impedance was lower than hypopharynx impedance in all test conditions. Impedance levels were significantly reduced in both locations during spirometry sets and recovery period, suggestive of increased liquid refluxate throughout the study.

Impedance levels were lowest during Spirometry Set 2. In the pan-esophagus, Set 2 impedance was significantly lower than impedance measured at baseline, recovery period and Set 1.

**Table 1. Patient characteristics and results from esophageal testing.**

| Clinical data | N = 24 |
|---|---|
| Female | 18 (75%) |
| Age [mean (SD)] (y) | 54 (±14.7) |
| No history of respiratory disorders | 20 (83%) |
| History of asthma (treated/non-symptomatic) | 4 (17%) |
| **Medications that affect esophageal function** | |
| Nitrates | 1 (4.2%) |
| Calcium channel blockers | 4 (17%) |
| **High-resolution impedance manometry results** | |
| No hiatal hernia | 15 (62%) |
| Hiatal hernia [median (IQR)] (cm) | 2.4 (1.6–2.8) |
| Normal high-resolution impedance manometry | 10 (42%) |
| Weak resting lower esophageal sphincter tone | 5 (21%) |
| Ineffective esophageal motility | 5 (21%) |
| Esophagogastric junction outflow obstruction | 1 (4.2%) |
| **Ambulatory 24 hr pH study conducted** | 17 (71%) |
| DeMeester Score [median (IQR)] | 37.7 (4.1–104.5) |

SD, standard deviation; IRQ, interquartile range

**Table 2. Impedance values at baseline and during spirometry study (n = 22).**

| Location | Mean (kΩ) ±SD | | | |
| --- | --- | --- | --- | --- |
| | Baseline (kΩ) | Spirometry Set 1 (kΩ) | Recovery Period (kΩ) | Spirometry Set 2 (kΩ) |
| Hypopharynx | 3.59 ±2.20 | 2.35 ±1.68[*] | 2.64 ±1.99[*] | 2.28 ±1.75[*] |
| Pan-esophagus | 2.52 ±0.74 | 1.57 ±0.59[*] | 1.60 ±0.57[*] | 1.37 ±0.67[*^#] |

[*] Paired t-test against Baseline $p<0.001$

[^] Paired t-test against Spirometry Set 1 $p<0.01$

[#] Paired t-test against Recovery Period $p<0.05$

The recovery period shows a trend towards restoring impedance to baseline levels but this was not statistically in the esophagus or above the UES.

## Manometry data

Pan-esophageal pressurization dynamics were noted during FVC maneuvers and resting period (**Fig 1**). At FVC, inspiration to TLC is characterized by diaphragmatic contraction and esophageal extension, leading to negative trans-esophageal pressurization (peak value -21.4 ±7.58 mmHg) relative to gastric pressure. Expiration to FVC generated large positive gastric pressures that prorogated back into the esophagus (peak value 122.3 ±31.3 mmHg). The LES and UES were hypertensive for the duration of the expiration maneuver (mean 217 ±71.7 mmHg and 258.3 ±78.7 mmHg, respectively).

Comparison of LES pressures showed significantly ($p<0.0001$) lower LES resting pressure during the recovery period after the first set of spirometry (13.6 ±8.8 mmHg) compared with baseline LES resting pressure (22.5 ±13.3 mmHg) (**Fig 2**). Normal LES resting pressure range is between 12 to 43 mmHg [10]. There was no significant difference in UES resting pressures pre-spirometry (57.9 ±18.4 mmHg) and post-spirometry (56.2 ±20.2 mmHg).

## Spirometry data

Eight (8) patients were unable to produce three reproducible FVC attempts per set, due to discomfort from the catheter or experienced a coughing episode. The remaining 16 of 24 (66.7%) participants successfully produce two sets of spirometry with each containing at least 3 reproducible spirometry attempts (**Table 3**).

Comparison of these spirometry sets showed significant reductions in the mean $FEV_1$ and PEFR values in spirometry Set 2 compared with those spirometry Set 1. Similarly, the EDI values were significantly higher in the second spirometry set compared to Set 1 and suggested laryngotracheal stenosis.

## Discussion

This study unveils the mechanisms underlying spirometry induced GER, elucidating how intra-esophageal pressure fluctuations and weakened LES tone following FVC maneuvers contribute to reflux. This induced refluxate can have a detrimental effect on spirometry reproducibility.

Tiller and Simpson [11] were pioneers in measuring the peak esophageal pressure during expiration, documenting values of 102 ± 34 cmH2O (or 75 ± 25 mmHg). They proposed that the forceful intra-abdominal contractions generated during expiration might facilitate the upward propulsion of gastric content into the esophagus. In our study, we similarly observed substantial intra-esophageal pressures during FVC expiration, which were instrumental in

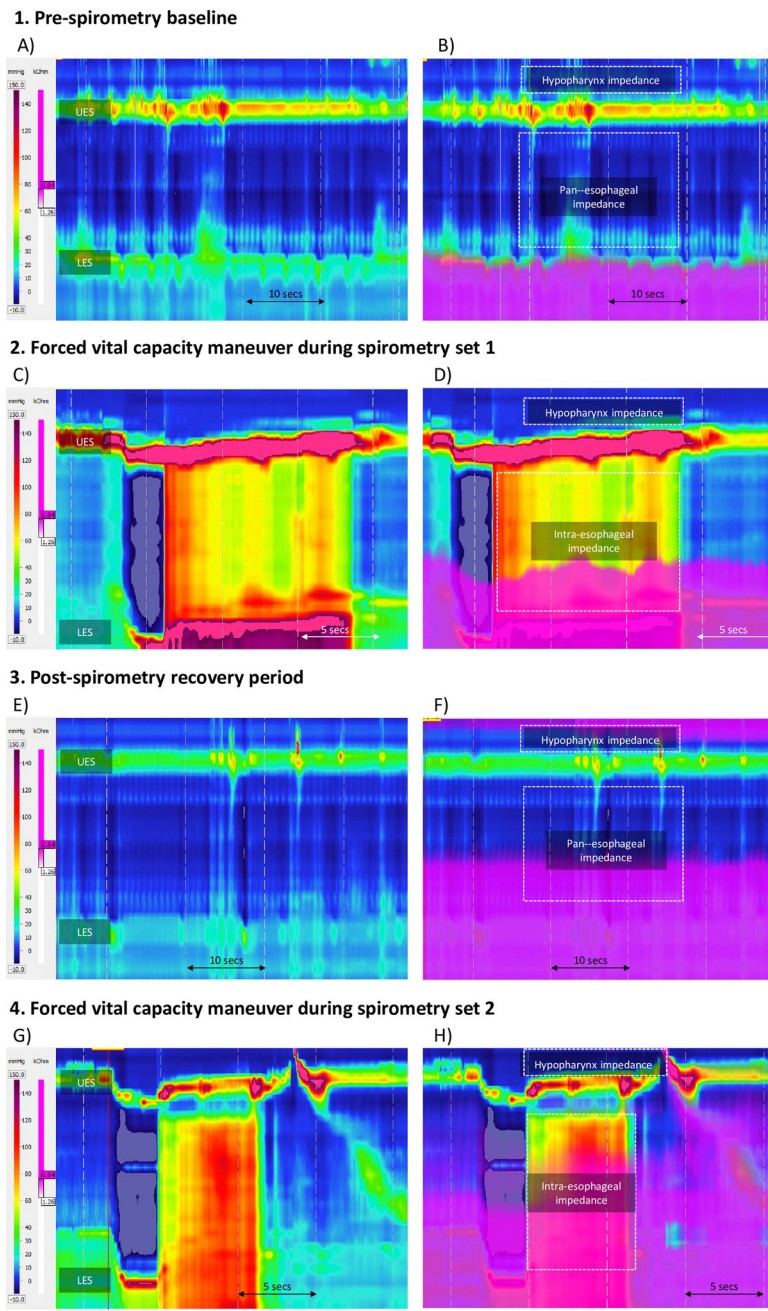

**Fig 1.** High-resolution impedance manometry topography plot (A, C, E, G); the intra-esophageal pressure is assessed in relation to time and distance. An impedance overlay (B, D, F, H); purple hue denotes fluid flow by impedance. Study time-points 1) Pre-spirometry esophageal pressures 2) A forced vital capacity maneuver during Spirometry Set 1. 3) Post-spirometry baseline during recovery period after Spirometry Set 1. 4) A forced vital capacity maneuver during Spirometry Set 2. UES, upper esophageal sphincter; LES, lower esophageal sphincter.

corroborating refluxate movement into both the esophagus and hypopharynx. These findings were validated using impedance fluid flow analysis. Notably, spirometry induced GER events were more pronounced in our current investigation than in our earlier study [2]. This discrepancy likely arises from enhanced sensitivity of impedance sensors, which can detect various

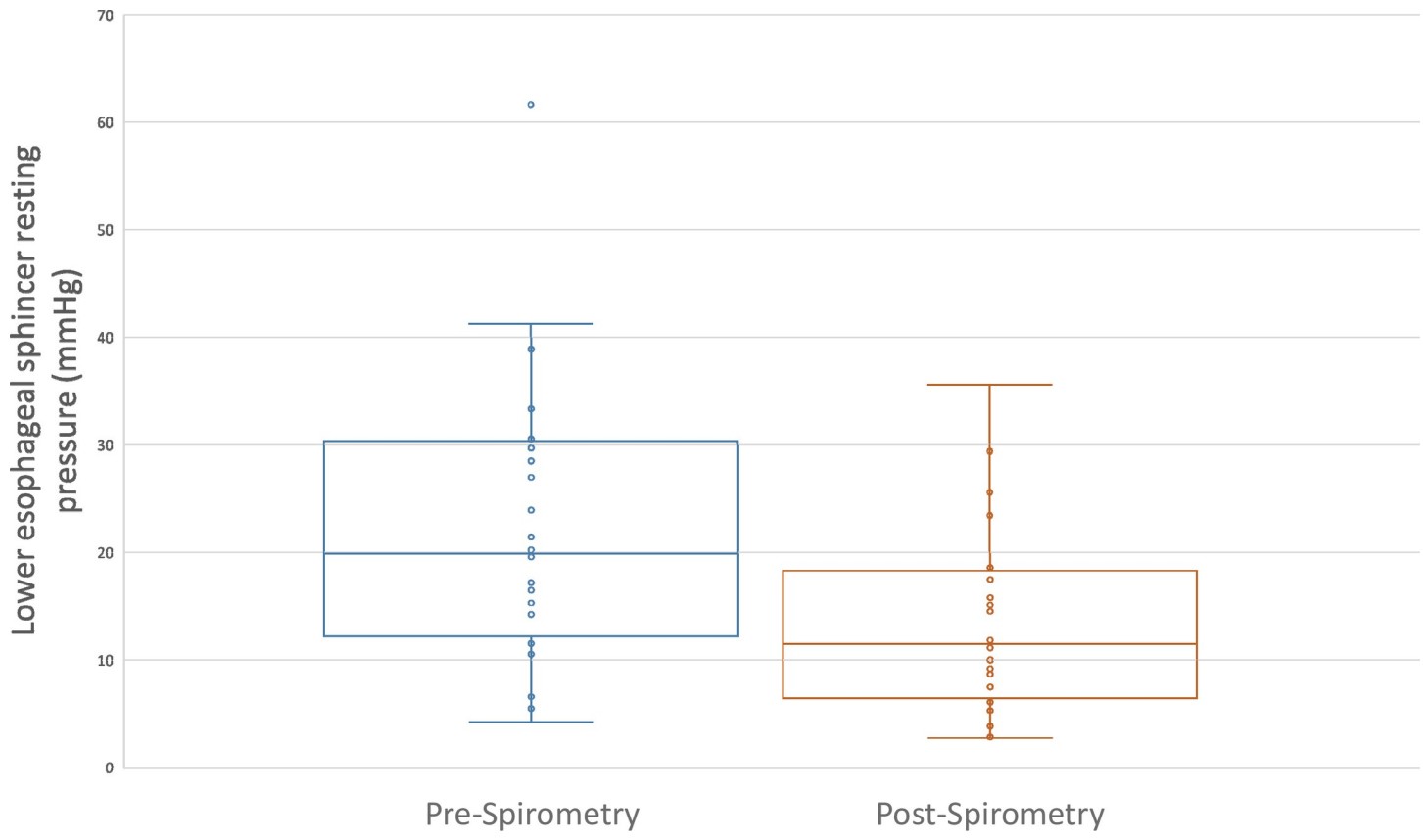

**Fig 2. Resting lower esophageal sphincter pressures (mmHg) before and after spirometry (n = 24).** Lower esophageal sphincter resting pressure was significantly lowered after spirometry ($p<0.0001$).

types of reflux (acidic, weakly acidic, and non-acidic reflux), in contrast to the ambulatory pH probe exclusively capturing acidic reflux in our previous research.

In healthy adults, activities that elevate intra-abdominal pressure, such as deep inspiration, forced expiration, trunk flexion, prompt the contraction of the right crus of the diaphragm. This muscular action augments pressure on the LES, fortifying the gastroesophageal barrier and thwarting GER. Nonetheless, physiologic stressors like strenuous exercise or respiratory ailments (e.g. coughing) can introduce fluctuations in intra-thoracic and esophagus pressures that surpass the resting LES pressure, thereby surmounting the protective barrier and leading

**Table 3. Analysis of spirometry set parameters (n = 16).**

| Parameter | mean ± SD | | | |
|---|---|---|---|---|
| | Spirometry Set 1 | Spirometry Set 2 | Difference | *p* value* |
| FVC (L) | 3.18 ±1.03 | 3.14 ±1.03 | 0.04 ±0.10 | 0.167 |
| $FEV_1$ (L) | 2.45 ±0.93 | 2.31 ±0.79 | 0.14 ±0.24 | 0.042 |
| PEFR (L/s) | 5.21 ±1.65 | 4.54 ±1.42 | 0.67 ±1.15 | 0.034 |
| EDI^ | 47.7 ±12.8 | 54.1 ±20.5 | 6.4 ±7.7 | 0.047 |

*Paired t-test

^EDI threshold >50 indicate anatomic laryngotracheal stenosis (sensitivity 95.9% and specificity 94.2%)

FVC: Forced vital capacity; FEV1: Forced expiratory volume in 1 second; PEFR: peak expiratory flow rate; EDI: Expiratory disproportion index.

to reflux [12, 13]. Our study contributes novel evidence that bolsters this concept, revealing a temporary reduction in LES resting tone subsequent to FVC maneuvers. This diminished LES tone predisposes patients to GER during the recovery period and subsequent FVC attempts. Although the exact cause of this LES weakening remains undetermined, we speculate that repetitive shifts in the intra-esophagus pressure gradient, particularly in a susceptible population, might contribute to LES fatigue. Alternatively, it is plausible that FVC may elevate the frequency of transient LES relaxation (TLESR), characterized by a sudden decline in the LES pressure independent of swallowing or secondary peristalsis. TLESRs typically arise from unconstrained air, such as belching, and play a physiological role in releasing stomach gas. Notably, the alterations in gastric pressure triggered by FVC could similarly incite TLESR responses. This presents a compelling avenue for further exploration of the mechanisms linking FVC maneuvers, LES function, and GER events.

Comparing parameters between spirometry Set 1 and Set 2, a noteworthy reduction in both $FEV_1$ and PEFR was observed in the latter set. While decreased $FEV_1$ and PEFR often arise from suboptimal patient effort, a an alternative explanation could be the reduction in large airway caliber [14]. A precedent study [9] successfully discriminated between cases of upper airway stenosis and non-stenosis using the EDI threshold above 50, yielding an impressive sensitivity of 95.9% and specificity of 94.2%. In our study cohort, the initial EDI value in Set 1 was 47.7, below the established threshold, which subsequently escalated to 54.1 by the time of spirometry Set 2. Potential ways that GER may contribute to transient constriction of the upper airways involve both direct (aspiration) or indirect (neutrally mediated) mechanisms [ref]. A disturbance of the normal protective mechanisms may allow direct contact of gastro-duodenal contents with the larynx or airway. This reflux may cause symptoms by irritation directly, or reflux may stimulate a vagal reflux arc producing cough and/or laryngospasm [15, 16]. Given the absence of underlying lung function disorders among participants, any effects induced by reflux-triggered laryngospasm remained ephemeral and had minimal repercussion on lung function. This aligns with the self-limiting nature of reflux-induced laryngospasm in the context of our study cohort.

Our study has inherent limitations that warrant consideration. Standard HRIM procedure may induce discomfort and stress in certain patients, potentially influencing their participation. Although there is currently no documented data regarding the impact of intubated naso-gastric catheters on the individual spirometry performance, it is plausible that their presence could exert adverse effects, potentially masking the true extent of participant's lung capabilities. Additionally, our patient cohort was deliberately selected to heighten the likelihood reflux events during spirometry, which may inadvertently limit the broader generalization of our findings, especially in the context of lung function assessment. Notably, the inference of gastric refluxate entry into the laryngotracheal region was derived from changes in the EDI rather than direct observation through manometry. To address these limitations, future investigations will strive to explore reflux dynamics in a population presenting respiratory symptoms. Utilizing non-invasive imaging technique, such as gastric labelling fluoroscopy could shed light on fluid movement within the laryngotracheal region, providing a more comprehensive understanding of the interplay between reflux and pulmonary function.

The implications of our study hold considerable significance, particularly in light of the widespread utilization of spirometry in routine lung function clinics. Consequently, our findings underscore the importance of respiratory physicians to incorporate a comprehensive assessment of patient history, encompassing reflux-related factors and ongoing reflux activity, during the interpreting spirometry results. This proactive approach serves to account for the potential influence of GER on pulmonary outcomes. Conversely, it is noteworthy that

anomalous signals in $FEV_1$ and PEF measurements during pulmonary function testing might serve as insightful indicators of concealed reflux activity.

In conclusion, the execution of FVC maneuvers during spirometry appears to trigger GER by inducing distinct intra-esophageal pressure differentials, followed by a transient weakening of the LES. Understanding the mechanisms of spirometry induced reflux may explain the origins of variability in spirometry outcomes, a variability that may be partially autonomous of patient effort. This holds particular relevance for individuals predisposed to GER.

## Supporting information

**S1 Checklist. STROBE statement—checklist of items that should be included in reports of** *cohort studies.*
(DOCX)

## Author Contributions

**Conceptualization:** John D. Brannan, Vincent Ho.

**Data curation:** Matthew Xu, John D. Brannan, Jerry Zhou.

**Formal analysis:** Matthew Xu, John D. Brannan, Jerry Zhou.

**Funding acquisition:** Jerry Zhou.

**Investigation:** Matthew Xu, Vincent Ho, Jerry Zhou.

**Methodology:** Matthew Xu, Vincent Ho.

**Project administration:** Vincent Ho, Jerry Zhou.

**Resources:** John D. Brannan, Jerry Zhou.

**Software:** John D. Brannan, Jerry Zhou.

**Supervision:** John D. Brannan, Vincent Ho, Jerry Zhou.

**Validation:** Matthew Xu, John D. Brannan, Jerry Zhou.

**Visualization:** Matthew Xu, Jerry Zhou.

**Writing – original draft:** John D. Brannan, Jerry Zhou.

**Writing – review & editing:** Matthew Xu, John D. Brannan, Vincent Ho, Jerry Zhou.

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
