## [Decision Letter · Decision Letter 0]

2 Aug 2023

PONE-D-23-07360Mechanism of spirometry associated gastro-esophageal reflux in subjects undergoing esophageal assessmentPLOS ONE

Dear Dr. Zhou,

Thank you for submitting your manuscript to PLOS ONE. After careful consideration, we feel that it has merit but does not fully meet PLOS ONE’s publication criteria as it currently stands. Therefore, we invite you to submit a revised version of the manuscript that addresses the points raised during the review process.

After carefully considering the reviewers' comments, it is evident that a major revision of the manuscript is necessary to improve the overall quality and significance of the work. The reviewers' input has provided valuable insights that will undoubtedly enhance the impact of our study.

We look forward to receiving your revised manuscript.

Kind regards,

Sampath Kumar Amaravadi, Ph.D

Academic Editor

PLOS ONE

Journal Requirements:

Reviewers' comments:

Reviewer's Responses to Questions

**Comments to the Author**

1. Is the manuscript technically sound, and do the data support the conclusions?

Reviewer #1: Yes

Reviewer #2: Yes

2. Has the statistical analysis been performed appropriately and rigorously? 

Reviewer #1: Yes

Reviewer #2: Yes

3. Have the authors made all data underlying the findings in their manuscript fully available?

Reviewer #1: Yes

Reviewer #2: Yes

4. Is the manuscript presented in an intelligible fashion and written in standard English?

Reviewer #1: No

Reviewer #2: Yes

5. Review Comments to the Author

Reviewer #1: Congratulations to the authors on a describing a novel phenomenon. While the paper is intriguing there are a few shortcomings:

1) the authors included patients with GERD defined as AET > 5.5%. However, as per the Lyon consensus documents AET> 6% was defined as GERD. Why did the authors select 5.5%. vs 6% ?

2) Along the same lines, DeMeester score only uses acidic events but not weakly acid and non acidic reflux events. The authors selected patients with DeMeester >14.72 but might have missed a population of patients with pre-existing non acidic and weakly acidic GER.

3) Did the authors screen patients for medications that could relax LES (e.g nitrates, calcium channel blocker, b2 agonist, benzodiazepines) ? This data needs to be included.

4) the authors also don't mention if any of the patients had underlying lung disease. Patients with restrictive and obstructive lung disease have some degree of GER due to abnormal intra-thoracic pressures. They should include this data in the manuscript.

5) in the Discussion section, line 185, the authors mention Tiller and Simpson first described peak esophageal pressure during FVC. This needs to be re-worded as "peak esophageal pressure during expiration at 102..." for better understanding.

6) in the Discussion section, paragraph 4, the authors infer the decline in FEV1 and PEF due to SGS based on EDI values. However, this cannot be a plausible explanation as none of these patients had SGS prior (based on Set 1 PFTs) and SGS develops over a long period of time. An alternative explanation needs to be mentioned. A laryngeal spasm due to reflux could be used to explain the drop in FEV1 and PEFR.

7) the authors proof read the manuscript and correct grammatical errors.

Reviewer #2: I appreciate your time, work and efforts while conducting this study but, in general, I feel the manuscript requires substantial revision. The manuscript was interesting to read. Found fruitful in its way and a good area to explore. The attachment has the comments inbuild to be reviewed upon.

6. PLOS authors have the option to publish the peer review history of their article (what does this mean?). If published, this will include your full peer review and any attached files.

Reviewer #1: No

Reviewer #2: **Yes: **Dr. Esha Arora

---

## [Author Response · Author response to Decision Letter 0]

22 Aug 2023

We would like to thank the reviewers for providing their comprehensive and well thought-out feedback. We have addressed their comments below: 

Reviewer #1: Congratulations to the authors on a describing a novel phenomenon. While the paper is intriguing there are a few shortcomings:

We would like to thank the reviewer’s motivating comment on the value of this study. 

1) the authors included patients with GERD defined as AET > 5.5%. However, as per the Lyon consensus documents AET> 6% was defined as GERD. Why did the authors select 5.5%. vs 6%?

Yes the reviewer is correct in that the Lyon consensus of more than 6% provides conclusive evidence of reflux. The 5.5% cut-off was used from work done by Pandolfino and colleagues back in 2003. Since they were pioneers with HRM and Digitrappers (Medtronic system) the Digitrapper has used 5.5% as the cut-off. That reflects in the accompanying software, which had not received a recent update to the software since 2018 which is when the Lyon Consensus was published. 

A re-analysis of patients with AET >6% was conducted and found that it did not change the total number of GORD patients in this study (n=17, 71%). The updated AET >6% criteria and reference to Lyon consensus has been amended in the manuscript [Line 37-38]. 

2) Along the same lines, DeMeester score only uses acidic events but not weakly acid and non acidic reflux events. The authors selected patients with DeMeester >14.72 but might have missed a population of patients with pre-existing non acidic and weakly acidic GER.

Impedance-pH monitoring allowed us to identify non-acidic or weakly acidic refluxate. There was no significant non-acidic or weakly acidic GER identified in our patient cohort. We have updated the results to address this cohort [Line 111-112].

3) Did the authors screen patients for medications that could relax LES (e.g nitrates, calcium channel blocker, b2 agonist, benzodiazepines)? This data needs to be included.

Medications that may relax LES or modulate esophageal function taken prior to the study was recorded and has been updated in Table 1. 

4) the authors also don't mention if any of the patients had underlying lung disease. Patients with restrictive and obstructive lung disease have some degree of GER due to abnormal intra-thoracic pressures. They should include this data in the manuscript.

Patients were screening for an existing lung disease diagnosis. Four patients did have non-symptomatic asthma at time of the procedure. No other lung diseases were identified. This has been amended in the results [Line 109-110]. 

5) in the Discussion section, line 185, the authors mention Tiller and Simpson first described peak esophageal pressure during FVC. This needs to be re-worded as "peak esophageal pressure during expiration at 102..." for better understanding.

This has been amended in the Discussion [Line 156].

6) in the Discussion section, paragraph 4, the authors infer the decline in FEV1 and PEF due to SGS based on EDI values. However, this cannot be a plausible explanation as none of these patients had SGS prior (based on Set 1 PFTs) and SGS develops over a long period of time. An alternative explanation needs to be mentioned. A laryngeal spasm due to reflux could be used to explain the drop in FEV1 and PEFR.

This has been updated in the text. Laryngospasm is a likely cause of transient constriction of the upper airway. Paragraph updated to better explain the mechanisms of upper airway reduction [Line 185-199]. 

7) the authors proof read the manuscript and correct grammatical errors.

The manuscript was reviewed and grammatical and spelling errors were corrected. 

Reviewer #2: I appreciate your time, work and efforts while conducting this study but, in general, I feel the manuscript requires substantial revision. The manuscript was interesting to read. Found fruitful in its way and a good area to explore. The attachment has the comments in build to be reviewed upon.

We would like to thank the reviewer’s feedback and improvements to the manuscript. The spelling and grammatical errors have been amended in the manuscript. Of particular note, is the addition of instrument validity/reproducibility for the Spiroscout and linked reference [Line 71-72].

---

## [Editor Report · Decision Letter 1]

30 Aug 2023

Mechanism of spirometry associated gastro-esophageal reflux in individuals undergoing esophageal assessment

PONE-D-23-07360R1

Dear Dr. Zhou,

We’re pleased to inform you that your manuscript has been judged scientifically suitable for publication and will be formally accepted for publication once it meets all outstanding technical requirements.

Kind regards,

Sampath Kumar Amaravadi, Ph.D

Academic Editor

PLOS ONE
---

## [Editor Report · Acceptance letter]

4 Sep 2023

PONE-D-23-07360R1 

Mechanism of spirometry associated gastro-esophageal reflux in individuals undergoing esophageal assessment 

Dear Dr. Zhou:

I'm pleased to inform you that your manuscript has been deemed suitable for publication in PLOS ONE. Congratulations! Your manuscript is now with our production department. 

Kind regards, 

on behalf of

Dr Sampath Kumar Amaravadi 

Academic Editor

PLOS ONE